# Coffee Consumption during the COVID Pandemic in a Portuguese Sample: An Online Exploratory Study

**DOI:** 10.3390/foods12020421

**Published:** 2023-01-16

**Authors:** Patrícia Batista, Anabela Afonso, Maria João Monteiro, Manuela Pintado, Patrícia Oliveira-Silva

**Affiliations:** 1Human Neurobehavioral Laboratory, Research Centre for Human Development, Universidade Católica Portuguesa, 4169-005 Porto, Portugal; 2Centro de Biotecnologia e Química Fina—Laboratório Associado, Escola Superior de Biotecnologia, Universidade Católica Portuguesa, 4169-005 Porto, Portugal; 3Department of Mathematics, School of Sciences and Technology, University of Évora, 7004-516 Évora, Portugal; 4Center for Research in Mathematics and Applications (CIMA), Institute for Advanced Studies and Research, 7004-516 Évora, Portugal

**Keywords:** coffee, consumption, COVID-19, telework, family

## Abstract

Background: Coffee is one of the most consumed beverages worldwide and is part of the Portuguese lifestyles. This study aims to describe coffee consumption during the COVID-19 pandemic, the change in consumption, the relation between work and familiar pressure during the COVID-19 pandemic and coffee consumption, and the reasons for this consumption pattern. Methods: This is a cross-sectional study conducted by an online questionnaire (*n* = 463) between March and June 2020. During the data collection phase, information about coffee consumption and socio-demographic characteristics were gathered. Results: All respondents were and are coffee consumers. The coffee average intake before the COVID-19 pandemic was 2.40 ± 0.84 cups of coffee per day, and the average consumption during the COVID-19 confinement was 2.68 ± 0.88 cups. Consumption increased during the COVID-19 pandemic, but a correlation between the consumption changes and the familiar or work pressure was not found. The general reasons for the increase in the coffee consumption were “social/cultural” (27%), “the search for energy “boost”” (22.9%), “to feel awake” (22.7%), “to deal with stress” (22.7%) and “the taste” (21.8%). Conclusions: The results suggest the ever-increasing popularity of coffee consumption. Respondents also highlighted that some situations make them more prone to consume coffee consumption, such as when they face stress and the need to control tiredness. The COVID pandemic depleted a change in behaviors.

## 1. Introduction

Coffee is one of the most consumed drinks worldwide and it has been essential in consumer culture in different countries since the mid-sixteenth century. According to the International Coffee Organization (ICO), in 2019/20, world coffee consumption was estimated at 168.86 million bags, 0.7% higher than in 2018/19.

This beverage consists mainly of caffeine (the best-known stimulant), with safe concentration limits (up to 400 mg/day in adults) recommended by the Food and Drug Administration (FDA). However, its impact changes from person to person, dependent on the physicochemical properties of the product, age, and dependent genetic and environmental interactions [1,2,3]. After consumption, caffeine is rapidly absorbed in the small intestine and diffuses rapidly in other tissues. Its consumption can induce harmful effects on health, such as an increase in heart rate, blood pressure, motor activity, attentiveness, gastric secretion, diuresis, and temperature [1,4]. However, when consumed in safe dosages, it also has beneficial health effects [5]. While the ICO estimated in the past that global coffee consumption would increase, it also anticipated a change in an unknown direction depending on the impact of COVID-19 [6].

Portugal is known as a coffee consumer country [7]. This can be partially explained by the literature showing that culture and socializing acts are the most reported reason for the consumption ratio [8,9,10]. In addition, the stimulating effect that comes from the consumption of coffee is another reason, usually an attempt to ensure physical and intellectual performance and the effects of decreasing fatigue [9,11,12,13,14,15,16,17]. Other reasons are taste, health perception, mood, and stress events [9,18,19]. However, the Portuguese data showed that males reported higher consumption compared to females [10,20].

COVID-19 has fundamentally changed how people live and work worldwide. Consequently, behaviors and attitudes have been changed, whether at a personal, professional, or familiar level, and this influences the consumption of substances, drugs, alcohol, and others [21,22,23]. As suggested by Crosta and collaborators (2021), psychological factors underlying changes in consumer behavior during the pandemic, such as stress, anxiety, and fear, have received less attention in the literature, while economic factors (e.g., work situation, work pressure, remote working, work suspension, and unemployment), social factors (e.g., social isolation, mandatory confinement/quarantine, family absence and/or family pressure with dependent dependents, and absence of socializing with friends), and the infection by COVID-19 disease or other pathologies have been the most discussed aspects implicated in consumer behavior change during pandemics [24].

The need to perform day-to-day functions, and the requirement to obtain high intellectual performance on the work, can help to explain the increase in coffee consumption during a challenging moment. Another source of stress was unscheduled teleworking, which has affected work-family balance, as working at home with the presence of children and sometimes other dependents generated extra pressure. Children at home also represented that many families were faced with helping their children in online learning [25,26]. This professional situation conditioned general well-being. Additionally, the sudden shift to teleworking, their uptake and acceptance, lack of specific regulations on this matter, generated anxiety, underlying behavior change, and unknown psychosocial consequences altogether created a problematic situation for most families [25,27,28]. With a focus on the perception of remote work, Baert and collaborators (2020) found that women, compared to men, experienced a smaller negative effect of telework related to work-family conflicts, and a greater increase in work performance [27]. Equally stressful, unemployment or lay-off experiences contributed to increased anxiety, family pressure, financial burdens, and a significant boost in substance consumption.

Taking into account stressful experiences during the pandemic and the increase in substance use in Portugal [29], this study aimed to understand the impact of coffee consumption during this period. In particular, it is important to relate coffee consumption to individual work situation. One important question is whether the impact of coffee consumption during the COVID-19 pandemic is related to the work situation (i.e., remote working, suspension from work, and/or unemployment). Furthermore, due to the particularities of this “legal drug” consumption, it is necessary to know more about its consumption, its impact on drinkers, and the potential risk of developing a pattern of abusive consumption. In addition to these, other variables should be studied such as the impact of price and the type of coffee on its consumption, among others.

However, this exploratory study aimed to describe coffee consumption during the COVID-19 pandemic, the change in consumption, the relation between work and familiar pressure during the COVID-19 pandemic and coffee consumption, and the reasons for this consumption pattern.

## 2. Methods

### 2.1. Procedure

Data were collected between March and May 2020 (COVID-19 mandatory confinement in Portugal) in a convenience sample of Portuguese citizens recruited from the institutional social media channels with an online questionnaire. The questionnaire was validated by a pilot study including a focus group and a readjustment of the items. It comprised two sections: One focused on socio-demographic data and another with closed-ended questions about coffee consumption habits (before and during the COVID-19 confinement; e.g., *“Before confinement, on average, how many coffees per day (on a typical day) did you drink?”*), perception about health (e.g., *“Overall, how do you rate your period of confinement with respect to your well-being/mental health?”*, work situation (e.g., *What is your employment status during Social Isolation?”)*, and the general experience during COVID-19-related confinement, e.g., *“Does my current coffee consumption (during confinement) worry me?”*.

### 2.2. Participants

In this study, 463 respondents completed the questionnaire, and the majority (77.8%) were women. The age was well distributed, as 38% of the respondents are younger than or equal to the average age (32.43 years; maximum age 63 and minimum 18). Most people worked (50.8%) or studied (34.6%). During the confinement period, people co-lived with their household, and 53.3% had children in their household (see Table 1).

### 2.3. Statistical Analysis

The statistical analysis was done using the software R Statistics version 4.0.4 (The R Foundation, Vienna, Austria). Categorical data are reported as percentages. Logistic regression was used to identify the factors associated with the change in coffee consumption at home. The considered covariates were: Household, number of children, work situation before confinement, and work situation during confinement. Kendall’s correlation coefficient was calculated to measure the relationship between consumption reasons before COVID-19 confinement and during confinement. To compare the probability of consumption of each reason before and during COVID-19 confinement was used, the Wilcoxon signed rank test, or the sign test when the assumption of symmetry was violated.

## 3. Results

In this study, we first evaluated the profile of coffee consumption before COVID-19 confinement. The results showed a high intake prevalence of coffee (Figure 1).

The average consumption before COVID-19 confinement was 2.40 ± 0.84 cups and the average during COVID-19 consumption was 2.68 ± 0.878 cups. 29.8% of the respondents increased the number of cups of coffee they drank during COVID-19, and 12.5% decreased the consumption.

To analyze the change in caffeine consumption, a new variable named “change of consumption profile” was created based on the frequency of coffee consumption at home compared to the pre-pandemic consumption and the during pandemic consumption. The results showed that 90.7% of the respondents changed (i.e., decreased or increased) the consumption profile.

Related to the reasons for coffee consumption before and during the COVID-19 pandemic’s mandatory confinement, the majority of participants reported (the top-five reasons) social/cultural act (27%), energy “boost” (22.9%), to feel awake (22.7%), to deal with stress (22.7%), and like the taste (21.8%). The least quoted reason was “to help focus” (17.5%) and these results corroborated other studies [19].

As previously reported, most respondents (53.3%) were in isolation in the household with children, and it should be noted that most respondents (79.7%) were teleworking, as shown in Table 1. Regarding the logistic regression, it did not indicate a significant association between the change in the consumption profile and household (*p* = 0.780), number of children (*p* = 0.526), work situation before confinement (*p* = 0.736) and work situation after confinement (*p* = 0.618).

The results showed a change when comparing consumption reasons before and during COVID-19 mandatory confinement (Figure 2). A decrease in consumption importance based on social/cultural aspects and an increase due to the need “to deal with stress” can be observed.

The data obtained also suggest that the reasons evoked by the participants for caffeine consumption are little or not at all correlated with each other (all *r*_Kendall_ < 0.5), whether in the situation before (Figure 3a) or during confinement (Figure 3b).

Data analysis showed a perfect agreement (*r*_Kendall_ = 1) before and during COVID-19 confinement related to the degree of frequency reported for caffeine consumption regarding the reasons: To feel awake/combat drowsiness, to help you concentrate, because you like it and because you like the energy “boost”. There is also almost perfect agreement in the degree referred for the reasons: To accompany a cigarette and to help you with weight control (Table 2).

Comparing before and during COVID-19 confinement, there was a significant change in the results when participants were asked about consuming coffee “to help cope with stress/anxiety” (*Sign test Me*_Before_ = 3, *Me*_During_ = 4, S = 2, *p* < 0.001). These data showed an increase in the frequency of times respondents consumed coffee because of their level of stress/anxiety. Another change was related to consuming coffee “for social habit” (*Wilcoxon test* Me_Before_ = 4, Me_During_ = 2, V = 10174, *p* < 0.001), for which respondents showed a decrease during the confinement. The “powerful diuretic” was another reason reported in data analysis (*Sign test* Me_Before_ = 1, Me_During_ = 2, S = 2, *p* < 0.001), but reduced during COVID-19 confinement.

## 4. Discussion

In this study, the data analysis showed an increase in coffee consumption by Portuguese citizens during COVID-19 confinement. There may be several reasons to explain this change in coffee consumption, such as stress, anxiety, work situation, and family pressure.

The work situation pressure experienced in teleworking situations, which involved adaptation, conjugation of family life, and the pressure in attending to children’s needs (e.g., to accompany them in studies and to pay attention) could have been a factor that increased the consumption of coffee, in order to guarantee physical and intellectual performance [26,28,30]. This is not completely new since previous studies have already shown the relation between coffee consumption and work before COVID-19 [11,19,31,32]. However, the suspension of work, the layoff regime in which many workers were placed, and the high unemployment rates were extra factors generating more stress, anxiety, and fear about the future. This stressful experience potentially required new strategies to regulate mood and improve mental activity, which could have triggered greater demand for using stimulant substances [30,33,34,35].

On the one hand, when analysing the reasons for coffee consumption before and during COVID-19 pandemic mandatory confinement, the respondents reported several reasons, with a major incidence in social and psychological effects. Regarding the most striking reason for coffee consumption in the literature, opinions are divided. On the other one hand, some studies highlight the “social/cultural” reason, as well as that reported in this study [8,19,36]. Moreover, other studies support that the energy boost provided by coffee is the most important reason and consumers expect to improve alertness and higher physical and mental performance [8,9,18,37].

Regarding the consumption due to the “taste,” our results with a Portuguese sample were in a different direction when compared to other studies [8,19,38], such as Sousa, in which they reported the personal pleasure (enjoy the beverage, pleasant aroma/flavour) with the main consumption reason [19].

After knowing the consumption reasons, we tried to understand if there was a change in the consumption reasons before and during COVID-19 mandatory confinement. It was found that the reasons related to social/cultural aspects decreased during COVID-19 confinement and increased the reason related to the need “to deal with stress”. As expected, the levels of stress and anxiety during the confinement plays an important role in the respondents’ lives, affecting the search for exogenous to regulate emotions and mood. Other studies have showed the same, for instance with alcohol consumption [39]. As for the decrease in the reason for consumption being due to the social act, this is easy to understand since social contacts are compulsorily interrupted due to the imposition of confinement.

In this study, respondents reported that they consumed coffee most of the time “to deal with stress”. However, the source of stress was not possible to explore in this study. During the pandemic, the potential stress sources are many, including the fear of the virus, the anxiety of the disease, the stress of a teleworking regime in conditions of reconciliation with family life, the pressure of family support, and the stress from loss of income [40]. These are some of the questions that remain unclear and would be pertinent to address in a future study.

## 5. Conclusions

As expected, the pandemic experience triggered a change in behavior, and consequently, in coffee consumption pattern. The results showed an increase in coffee consumption in this Portuguese sample during the COVID-19 pandemic.

The COVID-19 mandatory confinement imposed a series of challenges, among them an increased stress/anxiety level as a consequence of many changes in the personal-professional balance. While we expected to find a significant correlation between coffee consumption and telework or family conditions, we were surprised to find that these were not reported as the main reasons to consume coffee during the confinement. Instead, the coffee consumption “to deal with stress”, even for those without children or other dependent relatives in their household, was clearly highlighted. More studies are needed in Portugal and in other cultures with a limited coffee tradition to complement our findings.

While this is an explorative study, the results found here are very relevant to fill in a gap in the literature about those consumers and their interests or reasons for consuming coffee. Three of the most important commitments in this field are to create strategies to increase the literacy around coffee’s health consequences, to approach myths related to the impact of this substance and provide support for those with an exacerbated consumption. Consumers may easily be focused exclusively on this substance’s functional features and benefits, ignoring the consequences of an unhealthy consumption pattern.

This study has also drawn attention to the pattern of caffeine consumption, not only in relation to health effects, but also regarding the change in consumption during the COVID-19 pandemic and the reasons for this consumption pattern.

## 6. Limitations and Future Research

This study may have some limitations, such as the data obtained from a convenience sample, which may limit the generalizability of the findings. Future studies may aim for samples with statistical representativeness and compare consumers from different countries. This cross-country perspective may provide a comparative viewpoint that will allow us to understand the consumption pattern better. Likewise, considering a more heterogeneous sample in terms of gender, age, social status, and cultural background will permit the design of a wide range of aims, such as exploring how growth and other sociodemographic variables influence coffee consumption. Additionally, this study’s exclusive use of self-report can be considered a limitation. While past consumption is a strong predictor of consumption intention in the future [41], it is also susceptible to some cognitive biases, such as memory and temporal and social desirability. Therefore, future studies must try to reproduce the current results by applying other protocols that are not exclusively dependent on self-reporting.

Furthermore, as stated above, different sources of stress can show specific relationship patterns with coffee consumption behavior, which is important to understand the coffee consumer profile. Finally, future studies should be carried out to investigate this consumption of coffee and strategies for promoting health, and preventing risky behavior should be considered, as well as the design of an intervention plan on risk groups.

## Figures and Tables

**Figure 1 foods-12-00421-f001:**
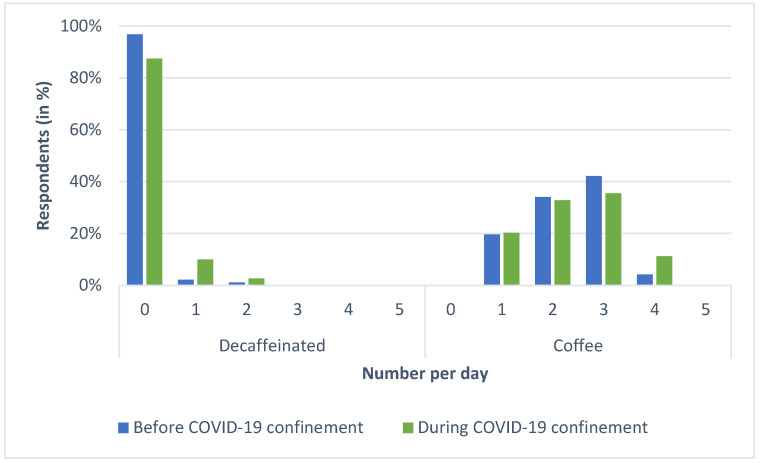
Decaffeinated (**left**) and coffee (**right**) change of consumption during COVID-19 confinement. Beverage consumption per day: 0 (without consumption), 1, 2, 3, 4, and 5 (five or more cups).

**Figure 2 foods-12-00421-f002:**
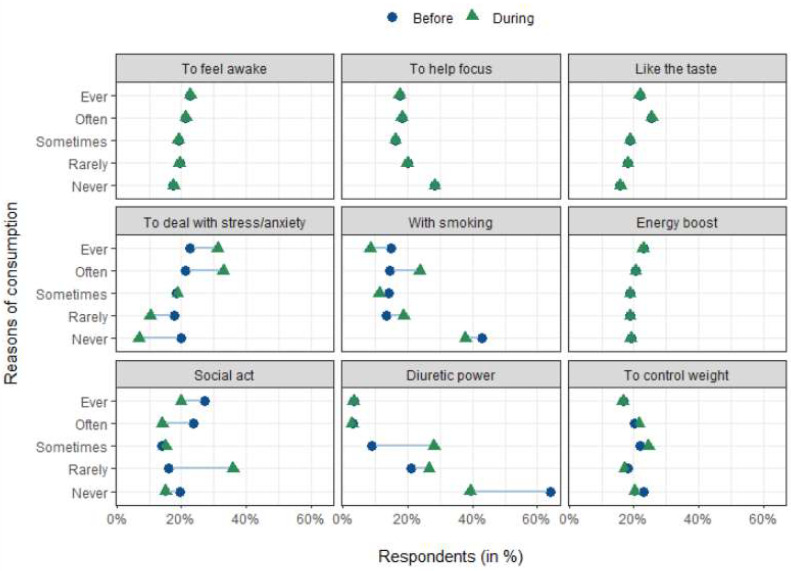
Reasons for coffee consumption considering before and during the COVID-19 pandemic mandatory confinement in a continous perspective.

**Figure 3 foods-12-00421-f003:**
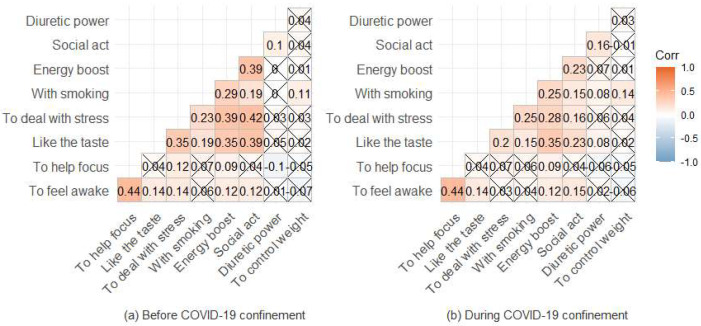
Kendall’s correlation coefficients for the reasons leading to coffee consumption (**a**) before and (**b**) during COVID-19 confinement (crossed values indicate non-significant correlations at 5%).

**Table 1 foods-12-00421-t001:** Sample characteristics.

Variables	Categories	N	%
Gender	Male	103	22.2
	Female	360	77.8
Household	Only	33	7.1
	With family (without child)	169	36.5
	With family (with child)	247	53.3
	Another situation	14	3
Number of Children	0	215	46.4
	1	175	37.8
	2	56	12.1
	3 or more	17	3.7
Work situation (before confinement)	Students	160	34.6
	Student worker	37	8.0
	Workers	235	50.8
	Unemployed	31	6.7
	Retired	0	0
Work situation (during confinement)	Work at workplace	2	0.4
	Telework	369	79.7
	Suspended work for family support	13	2.8
	Layoff situation	69	14.9
	Unemployed/retired/medical aid	10	2.2

**Table 2 foods-12-00421-t002:** Kendall’s correlation coefficients for each coffee-consumption reason appointed in this study, before and during COVID-19 confinement.

Coffee Consumption Reasons	*r* _Kendall_
To feel awake	1 ***
To help focus	1 ***
Like the taste	1 ***
To deal with stress	0.644 ***
To go with cigarrette	0.918 ***
Like the energy “boost”	1 ***
Social/cultural act	0.572 ***
A powerful diuretic	0.602 ***
To help weight control	0.966 ***

*Note*. *** *p* < 0.001.

## Data Availability

Data is contained within the article.

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
