# Peer review of "Coffee Consumption during the COVID Pandemic in a Portuguese Sample: An Online Exploratory Study"

_foods, 2023, doi:10.3390/foods12020421_

Round 1

Reviewer 1 Report

The paper seems very weak, especially in terms of methodology.

For example, where and how was the questionnaire administered?

The sample of respondents does not seem representative at all (males only 22.2% of the total?).

Author Response

Dear Reviewer,

We sincerely acknowledge the interest demonstrated in our work, the careful reviewers’ analysis, and the enriching suggestions/comments.

After looking keenly at the reviewers’ comments, we returned to the manuscript to undertake a series of modifications. After that, we believe the manuscript’s overall quality increased.

Thanks for the availability to reconsider a revised version of this manuscript.

The paper seems very weak, especially in terms of methodology.

Q1. For example, where and how was the questionnaire administered?

A1. We understand your concerns and now we try to clarify some of the methodology aspects.

Q2. The sample of respondents does not seem representative at all (males only 22.2% of the total?).

A2. We agree that the concern regarding gender’s balance (actually, this limitation is included in the article), but we assume that explaining this study represents a first approach (and the title reflects that including the designation “exploratory study”) the readers will frame our results based on this limitation. As said, the title was changed to clarify this situation (see page 1, line2-3).

Looking forward to hearing from you,

Patrícia Batista & co-authors

Reviewer 2 Report

Overall, the topic proposed by the manuscript is of interest to the readers and in line with the journal's aims and scope. The title can be improved for more attractiveness, while the Abstract, Introduction, Methods, Results, Discussion, Limitations and Future Research, and Conclusions and Suggestions are well structured, clear and concise. Literature is just enough.

The main purpose of this paper aims to describe coffee consumption during the COVID-19 pandemic, the change in consumption, the relationship between work and familiar pressure during the COVID-19 pandemic and coffee consumption, and the reasons for this consumption pattern .

The introduction and bibliographic sources seem to be sufficient.

The methodology seems to be good enough. Just add a paragraph with the analyzed sample and if it is representative.

The use of the software R Statistics is a plus (Logistic regression, Kendall's correlation and Wilcoxon signed rank test).

Everything is very well structured and easy for the reader to complete logically from Results and Discussion.

Figure 2, Figure 3 and Table 2 are easy to understand for the reader for a deeper understanding.

The use of bibliographic resources in the Discussion support the present analysis outlined by the authors. I consider that the results underlined in this material are sufficiently eloquent enough for this field in a continuous dynamic.

The conclusions are as relevant as possible and flow logically and intuitively from the analysis of the paper. Undoubtedly, as expected, the pandemic experience triggered a change in behavior and, consequently, in the coffee consumption pattern. Results showed an increase in coffee consumption in this Portuguese sample during the COVID-19 pandemic.

The limitations stated at the end of the article are relevant and suggest some justified limitations of the present research. I recommend moving them after the Conclusions chapter.

I think that, after minor corrections, the field literature in the reference section as well as the findings of both authors at this point are more than relevant enough for this paper.

Author Response

Dear Reviewer,

We sincerely acknowledge the interest demonstrated in our work, the careful reviewers’ analysis, and the enriching suggestions/comments.

After looking keenly at the reviewers’ comments, we returned to the manuscript to undertake a series of modifications. After that, we believe the manuscript’s overall quality increased.

Thanks for the availability to reconsider a revised version of this manuscript.

Q1. Overall, the topic proposed by the manuscript is of interest to the readers and in line with the journal's aims and scope. The title can be improved for more attractiveness, while the Abstract, Introduction, Methods, Results, Discussion, Limitations and Future Research, and Conclusions and Suggestions are well structured, clear and concise. Literature is just enough.

A1. Thank you very much for your careful reading and interest. And also, thank you for the suggested change of title (see page 1, line2-3).

The main purpose of this paper aims to describe coffee consumption during the COVID-19 pandemic, the change in consumption, the relationship between work and familiar pressure during the COVID-19 pandemic and coffee consumption, and the reasons for this consumption pattern.

The introduction and bibliographic sources seem to be sufficient.

Q2. The methodology seems to be good enough. Just add a paragraph with the analyzed sample and if it is representative.

A2. Thanks for your suggestion. We clarify this point (see page 3, line 98-99).

The use of the software R Statistics is a plus (Logistic regression, Kendall's correlation and Wilcoxon signed rank test).

Everything is very well structured and easy for the reader to complete logically from Results and Discussion.

Figure 2, Figure 3 and Table 2 are easy to understand for the reader for a deeper understanding.

The use of bibliographic resources in the Discussion support the present analysis outlined by the authors. I consider that the results underlined in this material are sufficiently eloquent enough for this field in a continuous dynamic.

The conclusions are as relevant as possible and flow logically and intuitively from the analysis of the paper. Undoubtedly, as expected, the pandemic experience triggered a change in behavior and, consequently, in the coffee consumption pattern. Results showed an increase in coffee consumption in this Portuguese sample during the COVID-19 pandemic.

Q3. The limitations stated at the end of the article are relevant and suggest some justified limitations of the present research. I recommend moving them after the Conclusions chapter.

A3. Thank you. We changed the place where limitations were discussed, accordingly (see page 2, lines 75-81).

I think that, after minor corrections, the field literature in the reference section as well as the findings of both authors at this point are more than relevant enough for this paper.

Once again, we thank you for your careful reading, interest, and suggestions for improving our article.

Looking forward to hearing from you,

Patrícia Batista & co-authors

Reviewer 3 Report

The research topic is very interesting and raises the awareness of the reader. However, I have some remarks about the research work:

I recommend that introduction needs to be revised by providing recent statistics on coffee consumption, especially the comparison before, during, and after COVID-19 in Portugal and around the world.

The novelty of the research, research gap, and research problem should be stated in the introduction. Also, there needs to be a brief introduction to how the adapted theoretical frameworks are supposed to help to understand the research problem and to fill the research gap. For example, why did you choose perception about health, work situation, and general experience? Similarly, why did you neglect other variables that have an impact on coffee consumption such as willingness to pay, personal preferences, or product attributes? Much research has indicated these motivations in recent literature. I strongly suggest improving the literature review.  

There is some remark about the methodology. The population and sample should be defined. What is your sampling technique? How did you reach the participants? Is your sample represent the population? More than two-thirds of your sample consists of females. Why?

How did you develop a new variable named “change of consumption profile”? What does it mean? How did you calculate this variable? If you categorize participants using the change of consumption amount before and during the confinement, what is your cut-off point? What about the significance?

The implications are very limited. Thus, I strongly recommend improving both theoretical and practical implications. Accordingly, I suggest highlighting the contribution of the paper to our knowledge about coffee consumption.

Author Response

Dear Reviewer,

We sincerely acknowledge the interest demonstrated in our work, the careful reviewers’ analysis, and the enriching suggestions/comments.

After looking keenly at the reviewers’ comments, we returned to the manuscript to undertake a series of modifications. After that, we believe the manuscript’s overall quality increased.

Thanks for the availability to reconsider a revised version of this manuscript.

The research topic is very interesting and raises the awareness of the reader. However, I have some remarks about the research work:

Q1. I recommend that introduction needs to be revised by providing recent statistics on coffee consumption, especially the comparison before, during, and after COVID-19 in Portugal and around the world.

A1. Thank you for your question. We would very much like to present these results, however, there is no information available what highlight even more the relevance of this study. We recently published an article about this issue (see page 2, lines 54-55).

Q2. The novelty of the research, research gap, and research problem should be stated in the introduction. Also, there needs to be a brief introduction to how the adapted theoretical frameworks are supposed to help to understand the research problem and to fill the research gap. For example, why did you choose perception about health, work situation, and general experience? Similarly, why did you neglect other variables that have an impact on coffee consumption such as willingness to pay, personal preferences, or product attributes? Much research has indicated these motivations in recent literature. I strongly suggest improving the literature review.  

A2. As frequently occur, this is study is just a part of a larger project exploring the impact of the COVID-19 pandemics in mental health and brain functioning. As such, we had no room to insert more variables or even restrictions regarding the format for the variables we chose. We believe the selected variables, although not perfect, can help us to better understand the perception of changes at the health and work situation aspects. As a preliminary study, it also aims to draw attention to this topic, and invite for further studies related to coffee and its impact on people's lives. Following your comment, we tried to clarify the aims (see page 3, lines 90-93).

Q3. There is some remark about the methodology. The population and sample should be defined. What is your sampling technique? How did you reach the participants? Is your sample represent the population? More than two-thirds of your sample consists of females. Why?

A3. The authors have corrected this section accordingly to the reviewers’ information, clarifying the methodology used (see page 3, line 98-99).

Regarding the representativeness, we understand this is a clear limitation and for this reason, it was included in the limitation section (see page 7, line271-279).

Q4. How did you develop a new variable named “change of consumption profile”? What does it mean? How did you calculate this variable? If you categorize participants using the change of consumption amount before and during the confinement, what is your cut-off point? What about the significance?

A4. Thanks for your question. We considered that there was a change in coffee consumption at home, i.e., when the category of the frequency of coffee consumption at home during COVID-19 confinement was different from that mentioned before COVID-19 confinement.". To clarify, in the statistical analysis methods we rephase the sentence: "Logistic regression was used to identify the factors associated with the change in coffee consumption at home, i.e, when the frequency of coffee at home during confinement was different from that mentioned before confinement.

Q5. The implications are very limited. Thus, I strongly recommend improving both theoretical and practical implications. Accordingly, I suggest highlighting the contribution of the paper to our knowledge about coffee consumption.

A5. Thanks for allowing us to clarify this point. We now developed the idea that this explorative study has drawn attention to the gap in the literature about those consumers and their interests/reasons for consuming coffee. On the other hand, it is important understand the coffee benefits, without ignoring the consequences of an unhealthy consumption pattern.

Looking forward to hearing from you,

Patrícia Batista & co-authors

Reviewer 4 Report

The authors raised an interesting issue related to potential lifestyle changes during a pandemic. However, both the aim of the study and the description of the study are not clear, which makes it very difficult to understand the results and, above all, to assess the scientific value of the work. In my opinion, it is necessary to complete the description of the methodology (research tool, survey process) and then present all the results of the analyses carried out. There is no analysis showing which reasons favored an increase in coffee drinking and which favored a decrease. Thus, I suggest to rethink the way the method and the results are presented in the manuscript.

Some very detailed comments:

The type of survey should be added in the title

Lines 58-65. Citation of sources is required.

Lines 81 - 86. This text needs rethinking. Nothing emerges from the introduction regarding caffeine addiction, also sentence “interest in knowing the impact of coffee consumption during the COVID-19 pandemic related to the work situation experienced” is not clear.

Lines 90 - 94. Information about the method and the questionnaire is not sufficient. There is no information on what was the selection of the study group, what questions were used (closed-ended questions information is insufficient to repeat the study by another researcher). Was the questionnaire validated?

Lines 125 -129. In addition to this information, the reader would like to know how many people have increased the number of cups of coffee they drink. The graph shows that the increase only applies to people drinking 4 cups.

Lines 127-128 To analyze the change in caffeine consumption, a new variable named “change of consumption profile” was created based on coffee consumption at home”.  The methodology lacks information on how this variable was created. How was the change defined? It appears from Figure 1 that the changes also consisted of a reduction in the amount of coffee consumed - this issue requires further analysis - this is, after all, one of the objectives of the study.

Lines 130- 132. This information does not refer to coffee drinking. It should be in the description of the study group or in a separate section describing the lifestyle of the subjects.

Lines 132-135. The results of this analysis should be presented in a table.

Lines 136 – 137. The sentence does not present results and therefore should not be included in this section. Can be used when formulating the objective.

Figure 2 . The way of the presentation of the results is nor correct. It seems that this is a continuous variable

Author Response

Dear Editor,

We sincerely acknowledge the interest demonstrated in our work, the careful reviewers’ analysis, and the enriching suggestions/comments.

After looking keenly at the reviewers’ comments, we returned to the manuscript to undertake a series of modifications. After that, we believe the manuscript’s overall quality increased.

Thanks for the availability to reconsider a revised version of this manuscript.

Looking forward to hearing from you,

Patrícia Batista & co-authors

The authors raised an interesting issue related to potential lifestyle changes during a pandemic. However, both the aim of the study and the description of the study are not clear, which makes it very difficult to understand the results and, above all, to assess the scientific value of the work. In my opinion, it is necessary to complete the description of the methodology (research tool, survey process) and then present all the results of the analyses carried out. There is no analysis showing which reasons favored an increase in coffee drinking and which favored a decrease. Thus, I suggest to rethink the way the method and the results are presented in the manuscript.

Some very detailed comments:

Q1. The type of survey should be added in the title

A1. The type of research was added in the title, by designation “… an online exploratory study”. The title was changed to clarify this situation (see page 1, line2-3).

Q2. Lines 58-65. Citation of sources is required.

A2. Thank you for your suggestion. We included a relevant reference (see page 2, line67).

Q3. Lines 81 - 86. This text needs rethinking. Nothing emerges from the introduction regarding caffeine addiction, also sentence “interest in knowing the impact of coffee consumption during the COVID-19 pandemic related to the work situation experienced” is not clear.

A3. We rewrite this sentence. We think now is clear. Thanks.

Q4. Lines 90 - 94. Information about the method and the questionnaire is not sufficient. There is no information on what was the selection of the study group, what questions were used (closed-ended questions information is insufficient to repeat the study by another researcher). Was the questionnaire validated?

A4. New information on the method has been added to clarify the methodology (see page 3, line99-100).

Q5. Lines 125 -129. In addition to this information, the reader would like to know how many people have increased the number of cups of coffee they drink. The graph shows that the increase only applies to people drinking 4 cups.

A5. This information was added and the consumption was more noticeable in people who were already heavy coffee drinkers (see page 4, line 156-158). In figure 1 the consumption of coffee (espress and double) was considered, and it was found that 12.5% people reduced the number of cups of coffee, and 29.8% increased it.

Q6. Lines 127-128 To analyze the change in caffeine consumption, a new variable named “change of consumption profile” was created based on coffee consumption at home”.  The methodology lacks information on how this variable was created. How was the change defined? It appears from Figure 1 that the changes also consisted of a reduction in the amount of coffee consumed - this issue requires further analysis - this is, after all, one of the objectives of the study.

A6. Thanks for your question. We considered that there was a change in coffee consumption at home, i.e., when the category of the frequency of coffee consumption at home during COVID-19 confinement was different from that mentioned before COVID-19 confinement.". To clarify, in the statistical analysis methods we rephase the sentence: 

To analyze the change in caffeine consumption, a new variable named “change of consumption profile” was created based on the frequency of coffee consumption at home comparing the pre-pandemic consumption and the during pandemic consumption. Results showed that 90.7% of the respondents changed (i.e., decreased or increased) the consumption profile.” (Specifically, 29 (6.3%) have reduced this frequency at home, and 391 (=84.4%) have increased the frequency).

And regarding the analysis:

"Logistic regression was used to identify the factors associated with the change in coffee consumption at home, i.e, when the frequency of coffee at home during confinement was different from that mentioned before confinement.”

Q7. Lines 130- 132. This information does not refer to coffee drinking. It should be in the description of the study group or in a separate section describing the lifestyle of the subjects.

A7. We have changed the location of the information, however this information should remain in the presentation of the results as it demonstrates that indicate no significant association between the change in the consumption profile and household, number of children, and other characteristics.

Q8. Lines 132-135. The results of this analysis should be presented in a table.

A8. We decided to describe the results because they were few and there was no need to include another table in the article.

Q9. Lines 136 – 137. The sentence does not present results and therefore should not be included in this section. Can be used when formulating the objective.

A9. We reformulated the sentence, to clarify the information.

Q10. Figure 2. The way of the presentation of the results is nor correct. It seems that this is a continuous variable.

A10. After discussing internally with the other authors, we decided to keep this table because from our perspective it represents well the progression in the consumption considering the “pre-pandemic pattern” with the “consumption during the confinement”. Although the measure itself is not a continuous variable, the approach/analysis denotes a continuous perspective of the data. We added a brief comment in the Figure 2 to clarify this issue.

Round 2

Reviewer 1 Report

Suggestions have been accepted so you can publish it.

Author Response

Dear reviewer,

Once again thank you very much for reading our article carefully and for all the comments. It was a great help to improve our work.

Reviewer 3 Report

The paper has not been significantly improved after the revision. There are many shortcomings, especially in terms of methodology and implications.  

Author Response

We are very sorry that the reviewer did not consider the changes in the last manuscript significant. Unfortunately, we believe the entire methodological approach used for this study was provided in the previous version of the manuscript. We understand that additional efforts must be made in order to better characterize the changes in the caffeine consumption pattern and the real impact of the pandemic. However, there is also evidence in the published literature suggesting that there is still a knowledge gap in the perception of caffeine consumption during the pandemic. Therefore, the main implication of this study is to increase awareness that may discourage excessive caffeine consumption habits acquired but also maintained during the confinements and the pandemics in general.

Reviewer 4 Report

Thank you for the changes in the manuscript, but I still have some doubts:

Q1. The type of survey should be added in the title

A1. The type of research was added in the title, by designation “… an online exploratory

study”. The title was changed to clarify this situation (see page 1, line2-3).

Ok. Why do you use “comma” in the title?

A3. We rewrite this sentence. We think now is clear. Thanks.

In my opinion, the amendment misses the point as it introduces information about the aim of the study and this is included in the next paragraph.

Q4. Lines 90 - 94. Information about the method and the questionnaire is not sufficient. There

is no information on what was the selection of the study group, what questions were used

(closed-ended questions information is insufficient to repeat the study by another

researcher). Was the questionnaire validated?

A4. New information on the method has been added to clarify the methodology (see page 3,

line99-100).

Ok, however I suggest to delete information about focus group as a way to validate the questionnaire.

Author Response

Thank you for the changes in the manuscript, but I still have some doubts:

Q1. The type of survey should be added in the title

A1. The type of research was added in the title, by designation “… an online exploratory

study”. The title was changed to clarify this situation (see page 1, line2-3).

Ok. Why do you use “comma” in the title?

A1 (new)

Thank you for your suggestion. We agree that the comma is not necessary (see page 1, line 2)

Q3. Lines 81 - 86. This text needs rethinking. Nothing emerges from the introduction regarding caffeine addiction, also sentence “interest in knowing the impact of coffee consumption during the COVID-19 pandemic related to the work situation experienced” is not clear.

A3. We rewrite this sentence. We think now is clear. Thanks.

In my opinion, the amendment misses the point as it introduces information about the aim of the study and this is included in the next paragraph.

A3 (new)

Thanks for emphasizing again the importance of enriching the literature section. Please see some changes in this section with the intention of providing additional relevant sources to our readers.

(see page 1, line 2)

Q4. Lines 90 - 94. Information about the method and the questionnaire is not sufficient. There

is no information on what was the selection of the study group, what questions were used

(closed-ended questions information is insufficient to repeat the study by another

researcher). Was the questionnaire validated?

A4. New information on the method has been added to clarify the methodology (see page 3,

line99-100).

Ok, however I suggest to delete information about focus group as a way to validate the

questionnaire.

A4 (new)

Thank you for your comment. We remove the information about focus group (see page 3, line 113).
